# Microclimate factors related to dengue virus burden clusters in two endemic towns of Mexico

**Johanna Tapias-Rivera**[1⊙]**, Ruth Aralí Martínez-Vega**[2⊙]**, Susana Román-Pérez**[3]**, Rene Santos-Luna**[3]**, Irma Yvonne Amaya-Larios**[4]**, Fredi Alexander Diaz-Quijano**[5]**, José Ramos-Castañeda**[6,7]*

**1** Maestría en Investigación en Enfermedades Infecciosas, Facultad de Ciencias Médicas y de la Salud, Instituto de Investigación Masira, Universidad de Santander, Bucaramanga, Santander, Colombia, **2** Escuela de Medicina, Facultad de Ciencias Médicas y de la Salud, Instituto de Investigación Masira, Universidad de Santander, Bucaramanga, Santander, Colombia, **3** Centro de Investigación en Evaluación y Encuestas, Instituto Nacional de Salud Pública, Cuernavaca, Morelos, México, **4** Centro Educativo de Humanidades, CEDHUM, Jiutepec, Morelos, México, **5** Department of Epidemiology–Laboratório de Inferência Causal em Epidemiologia (LINCE-USP), School of Public Health, University of São Paulo, São Paulo, Brazil, **6** Centro de Investigaciones Sobre Enfermedades Infecciosas, Instituto Nacional de Salud Pública, Cuernavaca, Morelos, México, **7** Facultad de Ciencias de la Salud, Universidad Anahuac, Ciudad de México, México

⊙ These authors contributed equally to this work.
* jramos@insp.mx

**Data Availability Statement:** The database with information from participants is unavailable since

## Abstract

In dengue-endemic areas, transmission control is limited by the difficulty of achieving sufficient coverage and sustainability of interventions. To maximize the effectiveness of interventions, areas with higher transmission could be identified and prioritized. The aim was to identify burden clusters of Dengue virus (DENV) infection and evaluate their association with microclimatic factors in two endemic towns from southern Mexico. Information from a prospective population cohort study (2·5 years of follow-up) was used, microclimatic variables were calculated from satellite information, and a cross-sectional design was conducted to evaluate the relationship between the outcome and microclimatic variables in the five surveys. Spatial clustering was observed in specific geographic areas at different periods. Both, land surface temperature (aPR 0·945; IC95% 0·895–0·996) and soil humidity (aPR 3·018; IC95% 1·013–8·994), were independently associated with DENV burden clusters. These findings can help health authorities design focused dengue surveillance and control activities in dengue endemic areas.

## Introduction

Dengue is an acute viral disease transmitted by *Aedes* mosquitoes [1]. The World Health Organization (WHO) reports 50 to 100 million cases yearly, of which 500,000 are severe and approximately 2·5% result in death [2]. From 1980 to 2022, Pan American Health Organization (PAHO)/WHO reported 34,553,761 dengue cases in the Americas region, including 458,547 (1·3%) severe cases and 15,644 deaths; in Mexico, in the same period, 2,330,045

this project is covert by the Mexican federal law on protection of personal data held by private parties (information about the law can be found in: https://privacyassociation.org/media/pdf/knowledge_center/Mexico_Federal_Data_Protection_Act_July2010.pdf). For specific inquiries about the data, interested subjects can contact the Chair of the Institutional Ethics Board (https://www.insp.mx/insp-cei.html; comunicacion@insp.mx). The dataset of climate variables was constructed as described in the material and methods section; the maps with the information necessary to generate the variables can be obtained from https://glovis.usgs.gov/app.

**Funding:** The author(s) received no specific funding for this work.

**Competing interests:** FAD-Q declares grants from the Brazilian National Council for Scientific and Technological Development – CNPq and Sanofi Pasteur not related to this work. JR-C declares grant support to develop the cohort from which data are taken and funded by Sanofi Pasteur DNG 22 (Dengue seroprevalence, neutralizing titers and incidence in an endemic population of Morelos State, Mexico); payment for expert testimony from Takeda and support for attending meetings and/or travel by Sanofi Pasteur not related to this work. RAM-V reports honoraria and travel expenses as a consultant for the project from which the primary data were obtained by the National Institute of Public Health. Mexico and funded by Sanofi Pasteur DGN 22. IYA-L reports honoraria as coordinator of the project from which the primary data were obtained by the National Institute of Public Health. Mexico and funded by Sanofi Pasteur DNG 22. JT-P, SR-P and RS-L have no conflict of interest to declare. This does not alter our adherence to PLOS ONE policies on sharing data and materials.

confirmed dengue cases were reported to PAHO/WHO, of which 111,244 (4·8%) were severe cases and 1,441 deaths [3]. The circulation of all four serotypes of the Dengue virus (DENV) has been reported in Mexico [3], making it a country with hyperendemic transmission [1]. In Mexico, as in many other parts of the world, passive surveillance for dengue is conducted [4]. Our group has studied dengue transmission in endemic towns in Mexico and found that around 60% of infections are asymptomatic, and an underreporting rate is over 90%, typical of dengue endemic areas [5, 6].

The WHO has considered the development of surveillance systems that take into account temporal and spatial heterogeneity as a priority in dengue prevention at the global level [7]. In addition, surveillance systems should consider house-to-house transmission [8], the effect of microclimate [9], and the immunity of people in the surveillance area [10]. In the same sense, understanding the impact of environmental variables on the burden of DENV infection is vital to focus surveillance and control strategies, and to establish policies that prevent increases in disease burden and transmission in endemic areas. Hahn and collaborators suggest that the introduction of geospatial assessments reveals possible associations of disease with environmental and social factors in specific areas that define the burden of disease [11].

Mexico has two vector-borne disease surveillance platforms which are not integrated; the epidemiological platform, which provides sociodemographic, clinical, and virological information on dengue cases [12, 13], and the entomological platform [14], which provides information on the number of eggs, some entomological indicators, surveillance at the health jurisdiction level on insecticide resistance, and on the presence of DENV in mosquitoes emerging from eggs collected from ovitrap surveillance [15]. None of these platforms integrate climatic information, among other reasons because the source of information is usually the patient's home or the home where the ovitrap is located, respectively. Although the systems provide useful information for the knowledge of dengue transmission, this information is not sufficient to manage prospectively the control actions of the viruses transmitted by *Aedes aegypti*.

In this context, geographic information systems and spatial analysis techniques applied to public health facilitate the detection of these areas of increased burden, making it possible to understand the phenomena that occur in the local geographic areas and thus contribute to the prevention, management, and control of endemic diseases such as dengue [16]. Therefore, the study aimed to assess the association between some microenvironmental determinants and the occurrence of burden clusters of DENV infection in a population-based cohort in two endemic towns in southern Mexico from 2014 to 2016. This study includes symptomatic and asymptomatic infections assessed by serology during 2·5 years of follow-up.

## Materials and methods

### Study area

Two dengue endemic towns in the state of Morelos, Mexico were selected taking into account the epidemiological information available up to the time of the study, which was provided by the Ministry of Health (SSA) of the federal government, where it was established that Morelos was the state with the most confirmed cases of dengue in Mexico. The first of them, Axochiapan (the urban area of the municipality of the same name) has approximately 18,659 inhabitants according to the 2015 INEGI estimated population; the temperature ranges between 13°C and 35°C. The second one, Tepalcingo (the urban area of the municipality of the same name) has approximately 12,895 inhabitants; the temperature varies between 19°C and 34°C (S1 Fig) [17–19]. Details of dengue transmission in the selected localities can be found in Martinez-Vega and collaborators, 2012 [20].

## Sources of information

A secondary analysis was conducted on data collected in a prospective population-based cohort study of people aged five years and older living in Tepalcingo and Axochiapan between 2014 and 2016. This cohort was assembled in 2011 and the objective of the primary study was to determine that dengue transmission occurs primarily in the peridomestic area of Index Cases in two Mexican endemic towns [5]. They were selected because they were dengue-endemic localities in Morelos with high incidence rates and similar population densities. In addition, the Ministry of Health has carried out a vector surveillance program since 2008.

In the second phase of the cohort study (2014–2016), the seroconversion dynamics were evaluated through the annual estimation of seroprevalence, and the association between previous serostatus and risk of incident DENV infection was determined [21].

This analysis included the subgroup of persons evaluated during the second phase of the cohort study who had participated during the first phase of the cohort and were DENV seronegative, had a recent DENV infection, or were random selected from seropositive participants without DENV recent infection (n = 461 of 862 included between August 2011 and March 2012) [5], or who were recruited in the second phase of the cohort and were DENV seronegative (n = 19 of 104 included between August and November 2014) (S2 Fig). Participants were assessed every six months on five occasions between August 2014 and November 2016 [21]. At each assessment, a survey was conducted, and a blood sample was taken for serological diagnosis of recent DENV infection [22]. The individually structured questionnaire included demographic variables and dengue symptoms. In addition, information on the house's characteristics was collected, and each dwelling's geographic location was obtained using a portable Global Positioning System (GPS, Garmin Ltd.).

## Recent DENV infection

Recent DENV infection was defined based on the seroprevalence status and the year when the survey was done (Table 1), using IgG Indirect ELISA test (Cat E-DEN 01 G), and Panbio® IgG and IgM capture ELISA tests (Cat No. E-DEN02G and E-DEN01M) following the manufacturer's indications of Panbio® [22]. This definition was based on the fact that no other flavivirus was circulating in Mexico after the re-emergence of DENV in the late 1970s until the introduction of Zika virus in 2015.

## Microclimate variables

The microclimate analysis was carried out using the method of Roman-Perez and collaborators [23]. Briefly, micro-climatic variables of soil humidity were calculated through the Tasseled Cap transformation for 2014–2016. The land surface temperature (LST) was computed with the Split Window algorithm with radiometric, emissivity, and atmospheric correction at a resolution of 30 meters [24, 25]. These micro-climatic variables were obtained from 19 satellite images from the Landsat 8 OLI-TIRS sensor from the United State Geological Survey (USGS) and the National Aeronautics and Space Administration (NASA) [26]. Averages of LST and soil humidity were calculated for each study area polygon and survey. The variable humidity was reclassified dichotomously for analyzing micro-climatic factors associated with DENV burden clusters (high humidity: high and very high = 1; very low, low, and medium = 0).

## Cluster and hotspot analysis

Burden clusters of recent DENV infection in both towns were analyzed in order to search for spatial clusters of the event in the studied areas. In addition, we identified recent DENV

**Table 1. Definition of recent confirmed DENV infection.**

| Survey | Date | Test | Definition of infection* |
|---|---|---|---|
| 1 | August to November 2014 | IgM capture ELISA and IgG capture ELISA | a. IgM positive.<br>b. IgM and IgG positives.<br>c. IgM negative and IgG positive, only if the ratio of Panbio units obtained in the measurement from August to November 2014 and those obtained between August 2011 and March 2012 were greater than 1.37 |
| 2 | February to May 2015 | IgG Indirect ELISA in seronegative participant | a. Seroconversion of IgG antibodies: negative in survey 1 and positive in survey 2 |
| | | IgM capture ELISA and IgG capture ELISA | b. IgM and/or IgG positive in survey 2, but without infection in survey 1 |
| 3 | August to November 2015** | IgG Indirect ELISA in seronegative participant | a. Seroconversion of IgG antibodies: negative in survey 2 and positive in survey 3 |
| | | IgM capture ELISA | b. IgM positive survey 3, but negative in survey 2 |
| 4 | February to May 2016** | IgG Indirect ELISA in seronegative participant | a. Seroconversion of IgG antibodies: negative in survey 3 and positive in survey 4 |
| | | IgM capture ELISA | b. IgM positive survey 4, but negative in survey 3 |
| 5 | August to November 2016** | IgG Indirect ELISA in seronegative participant | a. Seroconversion of IgG antibodies: negative in survey 4 and positive in survey 5 |
| | | IgM capture ELISA | b. IgM positive survey 5, but negative in survey 5 |

* Indeterminate results were considered negative. The indeterminate results were 1·0% in IgG Indirect ELISA, 1·4% in IgM capture ELISA, and 8·8% in IgG capture ELISA.

**The capture IgG test was not considered in these latter periods because it cross-reacts with the Zika virus, which started circulating in Mexico in November 2015.

infection hotspots, which allowed us to identify areas with statistically significant higher levels surrounded by other entities with high values, in Axochiapan and Tepalcingo, to understand the spatial patterns of recent DENV infection in both Mexican towns.

To determine the spatiotemporal variation of recent DENV infection, a polygon network was constructed on the point maps of the towns. The size of each polygon was 200 meters to the centroid. On this grid, study participants were located according their recent DENV infection status (infected/not infected, Table 1) in each of the five surveys. The DENV infection frequency was calculated for each polygon. Getis-Ord Gi* tool were used to identify "hot spots" with 99% confidence of the occurrence of recent DENV infection in the polygons [27]. In addition, the Moran's Statistic for autocorrelation analysis (ESRI—Cluster Anselin Moran's I) was used at a 95% confidence level to identify high burden clusters; each entity was analyzed within the context of neighboring entities located within the specific distance band (200 meters) using ArcGIS 10.6℞ software. SQL Server was used to create a Geodatabase, Bing maps were used, and satellite images of each town were obtained from Google Earth©. Data from participants from neighborhoods far from the urban area of Axochiapan were not included in the analysis because their inclusion would have resulted in some polygons having no neighbors and would have increased the number of zeros.

## Statistical analysis

We considered the dichotomous variable of the household belonging to a burden cluster of recent DENV infection in each survey for 2·5 years as the outcome, it was previously defined with the Moran's statistic with 99% reliability. This statistic allowed evaluating the spatial autocorrelation of the data set, which refers to the degree to which the values of the interest variable were similar or different at neighboring locations. That is, it allowed us to infer if similar values are grouped in space [28]. In addition, the exposures of interest were the LST and soil

humidity because these variables influence the microclimate and could modify the vector population. Also, household characteristics were evaluated as independent variables.

A univariable multilevel Poisson regression analysis was performed to calculate the crude Prevalence Ratio (PR), considering the household as the first level unit and the hexagon to which the household belonged as the second level unit. Subsequently, a multilevel Poisson regression was performed with the two microclimatic variables (LST and soil humidity), and this model was adjusted for the housing variables with a $p <0·20$ in the univariable analysis. An adjustment was made for the number of the survey (1 to 5) to estimate the total effect of the two exposures of interest (Environmental model). In addition, a multiple regression model was built, adjusting by some household characteristics, to determine the direct effects of the two climatic variables (Complete model). The analysis was performed using STATA 15·1®.

## Results

### Recent DENV infections and characteristics of the study population

Five surveys were conducted between August 2014 and November 2016, 480, 560, 534, 518, and 435 participants were evaluated between the first and the fifth surveys, respectively. In Axochiapan, 152 households were monitored in the first survey and 135 in the fifth survey, and in Tepalcingo, 86 and 83 homes were monitored, respectively. In all surveys, recent DENV infections were detected, 50% of these (n = 88) were observed at the first survey (August-November/2014), and 22% (n = 38) were seen at the last follow-up (August-November/2016). The frequency of recent DENV infections varied between 1·5% and 18% across the surveys (S2 Fig).

Demographic characteristics were similar in both towns, with 57·1% to 61% female and the most frequent occupations being student and housewife. Also, 50% of the participants were between 13 and 41 years old, with a median age of 24 (S1 Table). Regarding the availability and waste of water in the dwellings in all surveys, most households had drains connected to the public network (>90%), and more than 69% had a public water network inside the dwellings. About 24% of households are supplied with water from wells, and more than 56% do not have mosquito nets on access doors or windows (S2 Table).

### Recent DENV infections in clusters and hotspots

In Axochiapan, in surveys 1, 3, and 5 (August-November 2014, 2015 and 2016), 32 hotspots of recent DENV infection were observed predominantly in the southeastern area of the town. In Axochiapan surveys 2 and 4 (February-May 2015 and 2016), 14 hotspots were observed (S3 and S4 Figs). However, autocorrelation analysis only identified significant clusters of recent DENV infection in six polygons in three neighborhoods in survey 1 (Vista Hermosa to the northwest and El Carmen and El Progreso to the southeast, with infection frequencies between 10% and 100%) and four polygons of two neighborhoods in survey 3 (El Carmen and El Progreso with infection frequencies between 5·6% and 33·3%) (Figs 1 and 2, S3 Table). The hotspots identified in surveys 2, 4, and 5 in this town did not consolidate as clusters of the burden of DENV infection (S3–S5 Figs).

In Tepalcingo, in surveys 1 and 2 (August/2014 and May/2015), a high frequency of recent DENV infection was observed in the northwestern (11 hotspots) and southeastern (4 hotspots) areas in survey 1, while 4 and 5 hotspots were found in survey 2, respectively. However, autocorrelation analysis only identified significant clusters of infection in 1 polygon of each neighborhood in each survey (S6 and S7 Figs, S4 Table). On the other hand, in survey 3, although ten hotspots were identified, none were burden clusters (S8 Fig). Moreover, in survey 4 (February/2016), seven hotspots were observed in a single neighborhood (San Francisco), of which

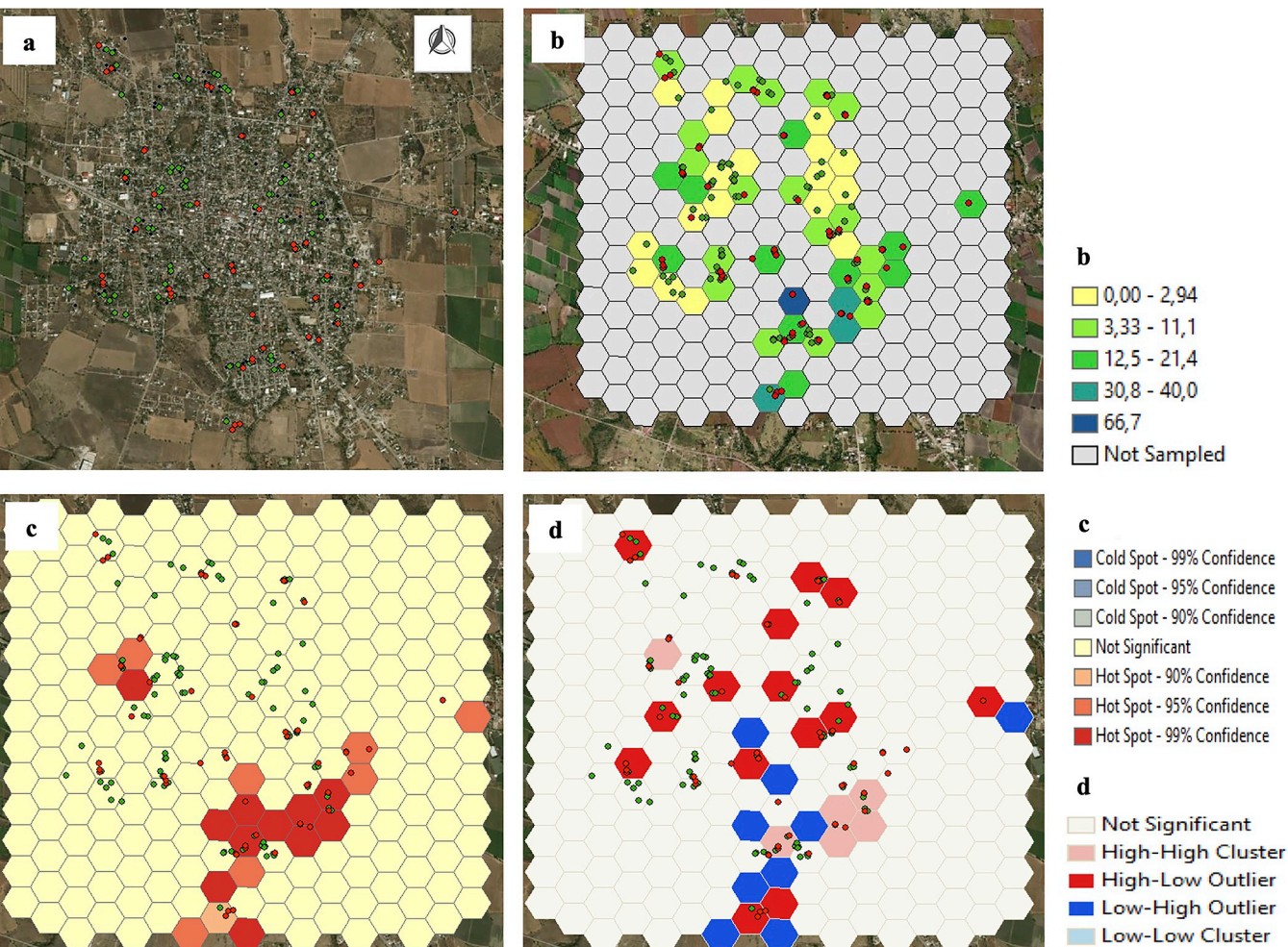

**Fig 1. Spatial distribution and burden clusters of recent DENV infections in Axochiapan, first survey.** a. Spatial distribution of the participants (Red dots: Positive for recent DENV infection; Green dots: Negative for recent DENV infection). b. Percentage of recent DENV infection. c. Hot Spots of recent DENV infection. d. Clusters (Autocorrelation, Anselin local Moran's I). Sources: Esri module of ArcGIS, DigitalGlobe, GeoEye, Earthstar Geographics, CNES/Airbus DS, USDA, USGS, AeroGRID, IGN, and the GIS User Community.

two were identified as burden clusters (S9 Fig), while in survey 5 (November/2016), 11 hot-spots were observed in four neighborhoods, of which only two were identified as burden clusters (one in each neighborhood, San Francisco and Palo Revuelto, S10 Fig).

In Axochiapan, 4 out of 6 polygons identified as burden clusters in survey 1 were also clusters in survey 3 (S3 Table). Although in Tepalcingo, only one polygon repeated the condition of being a cluster of recent DENV infection in the same season of the year (surveys 2 and 4), the other identified burden clusters were neighbors of the cluster detected in survey 1 (S7–S9 Figs, S4 Table). In addition, it is striking that most of the clusters are adjacent to the cemeteries of the towns (S11 and S12 Figs).

## Description of microclimate variables

Using remote sensing techniques, the micro-climatic variables of soil humidity and LST were calculated from satellite images. In Axochiapan, the highest soil humidity was observed in survey 1 (Fig 3A), while in surveys 2 and 4, low humidity predominated throughout the town

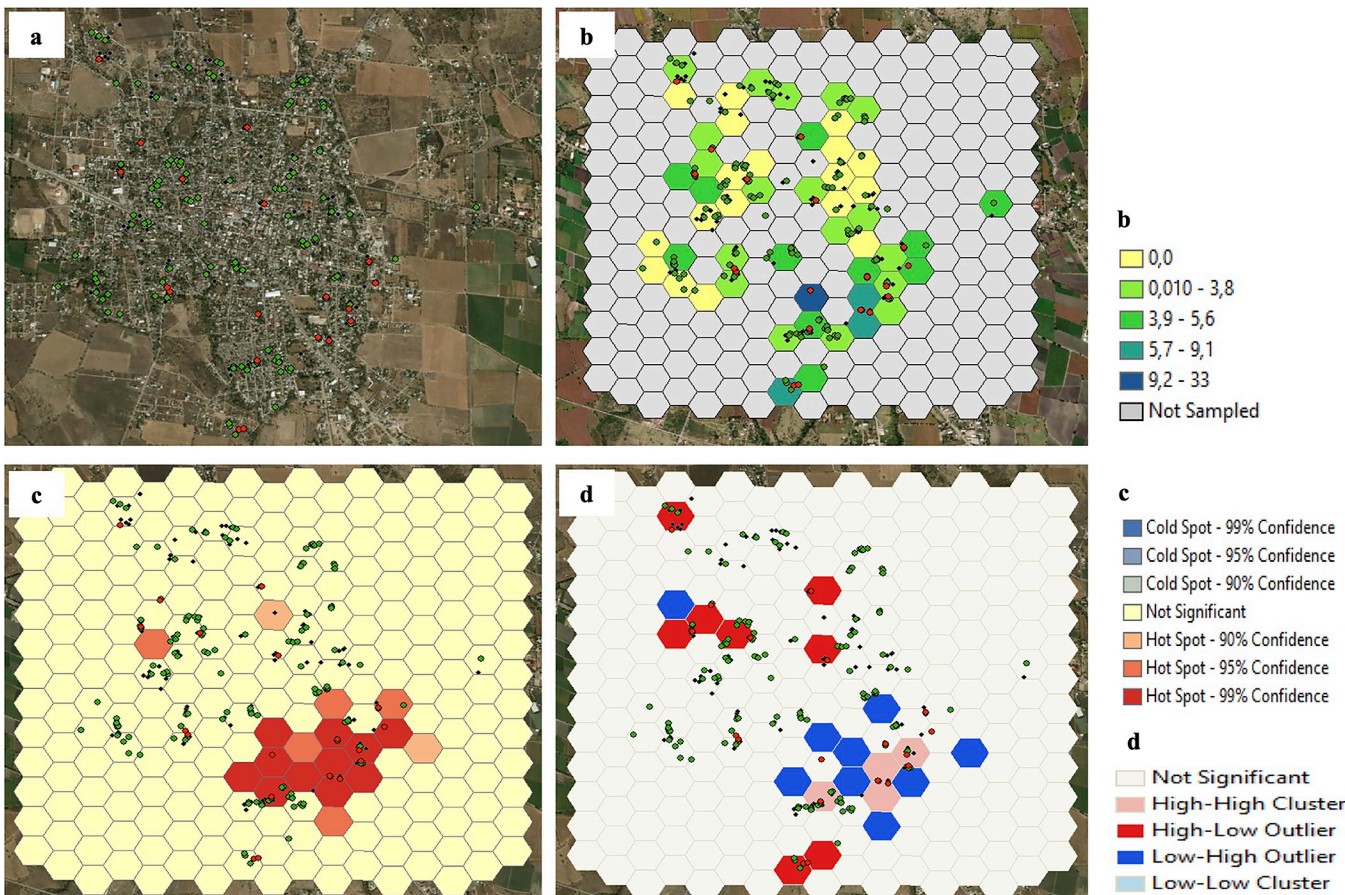

**Fig 2. Spatial distribution and burden clusters of recent DENV infections in Axochiapan, third survey.** a. Spatial distribution of the participants (Red dots: Positive for recent DENV infection; Green dots: Negative for recent DENV infection). b. Percentage of recent DENV infection. c. Hot Spots of recent DENV infection. d. Clusters (Autocorrelation, Anselin local Moran's I). Sources: Esri module of ArcGIS, DigitalGlobe, GeoEye, Earthstar Geographics, CNES/Airbus DS, USDA, USGS, AeroGRID, IGN, and the GIS User Community.

(S13 Fig). Although humidity was higher in the second half of the year (surveys 1, 3, and 5) compared to the first half of the year, a decrease in moisture was observed over the years (Fig 3A and 3B, and S13 Fig). Regarding the LST in Axochiapan, in the three surveys of the second half of the year, the temperature ranged between 32·8˚C and 41·5˚C, with a similar pattern in surveys 1 and 3, where the center of the town was warmer (Fig 3C and 3D). In contrast, in the surveys of the first half of the year (surveys 2 and 4), LST ranged from 41·2˚C to 53·8˚C. Still, a differential pattern was observed because the center of the town was warmer in survey 2, while this area was less warm in survey 4 (S14 Fig).

In Tepalcingo, survey 2 showed the lowest humidity of the period compared to survey 5, which was the wettest. In surveys 1 and 3 (second half of the year), the behavior was similar, with the lowest humidity in the center of the town. In addition, a significant difference was observed in the behavior of this variable in the first half of 2015 compared to 2016, with the first half being less humid (S15 Fig). Regarding the LST in Tepalcingo, in surveys 1 and 3, the LST ranged between 30·8˚C and 44·2˚C, with a similar pattern where the center of the town was warmer. In contrast, in the surveys of the first half of the year, the LST ranged between 32·7˚C and 46·3˚C, with a similar pattern where the center of the town was less warm than the periphery (S16 Fig). It is noteworthy that survey 5 had the lowest LST of the period (-1·6˚C to 30·2˚C), which coincides with the high humidity observed (S15 and S16 Figs).

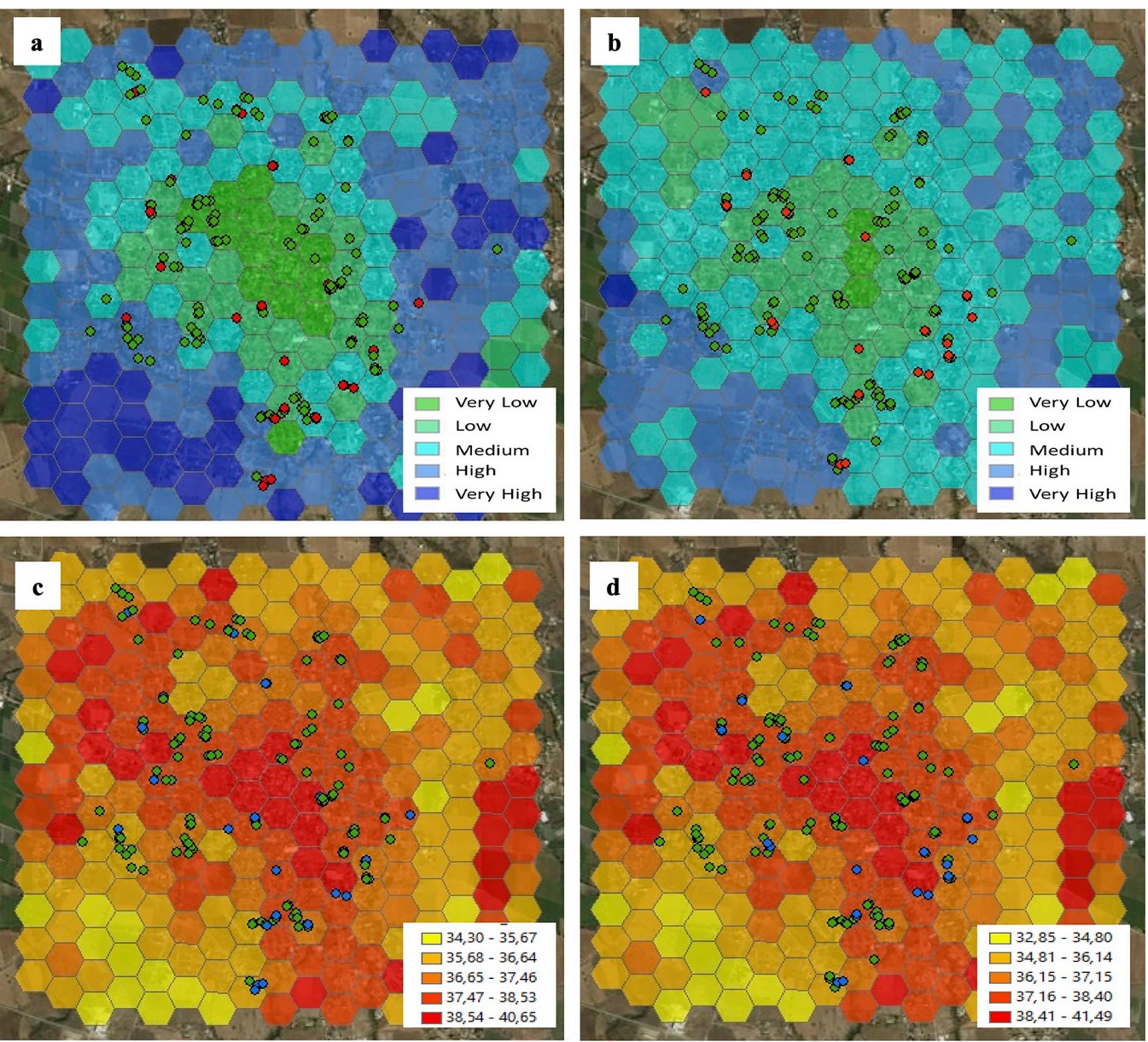

**Fig 3. Spatial distribution of soil humidity and land surface temperature in Axochiapan, first and third surveys.** Red or blue dots: Positive for recent DENV infection; Green dots: Negative for recent DENV infection. a. Soil humidity in survey 1. b. Soil humidity in survey 3. c. Land surface temperature in survey 1. d. Land surface temperature in survey 3. Sources: Esri module of ArcGIS, DigitalGlobe, GeoEye, Earthstar Geographics, CNES/Airbus DS, USDA, USGS, AeroGRID, IGN, and the GIS User Community.

## Microclimatic factors associated with clusters of recent DENV infections

The hexagon LST was negatively associated with households belonging to a DENV burden cluster, while high soil humidity had a positive association with this outcome (Table 2). These associations held even when adjusting for housing variables related to the outcome in the univariable analysis and for the survey time (Table 2 and S5 Table). To our knowledge, this is the first report demonstrating a point association between a burden cluster of DENV infections and climatic variables in a sublocal area. We will discuss this finding in more detail, but it should be noted that until now, the relationship between dengue and climatic factors has been

**Table 2. Microclimatic variables related to burden clusters of DENV infection.**

| Variable | n/n° Groups/ Hexagons | Univariate model PR (95% CI) | Environmental model aPR (CI95%) | Complete model[a] aPR (95% CI) | p |
|---|---|---|---|---|---|
| Land Surface temperature | 1153/266/84 | 0.929 (0.885–0.975) | 0.954 (0.909–1.002) | 0.945 (0.895–0.996) | 0.037 |
| High soil humidity | 1153/266/84 | 4.228 (1.731–10.324) | 3.348 (1.253–8.944) | 3.018 (1.013–8.994) | 0.047 |
| Number of surveys | | | 0.550 (0.433–0.698) | 0.542 (0.419–0.700) | <0.001 |

[a]The model was adjusted for the household variables: Poor housing maintenance, computer, refrigerator, piped water, and household infestation with larvae and pupae.
PR: Prevalence Ratio. aPR: Adjusted Prevalence Ratio.

studied based on case reports and at the relatively aggregated spatial scale of the city [29]. We demonstrate a relationship between DENV infections, including asymptomatic infections [21], at a spatial scale smaller than the neighborhood scale (200-meter polygons) that allows dengue surveillance in endemic populations and rationalization of vector control measures to increase their efficiency.

## Discussion

This spatial and temporal analysis showed burden clusters of recent DENV infection during two and a half years of follow-up in two endemic towns in Mexico that coincided with the same seasons of the year and with the very well know cyclical rate incidence describe in many studies in dengue endemic cities [30, 31]. Moreover, in 2014, more dengue cases were reported by the surveillance system in Morelos than in the following two years. In particular, Tepalcingo had the state's second-highest number of dengue cases. Furthermore, the study showed that burden clusters occurred in these dengue endemic areas repeatedly at localized points and specific periods. In addition, the household belonging to these DENV burden clusters was associated with the microclimatic variables of LST and soil humidity, independently of household characteristics.

Due to their number of inhabitants, Axochiapan (18,659) and Tepalcingo (12,895) are classified as rural-urban towns; they have more than 60% of the population living in poverty and the level of marginalization is high to very high. In addition, 30·1% of the population of Axochiapan and 46·2% of the population of Tepalcingo live in houses lacking basic services. In this study we determine that about 30% of the cohort participants' households lack piped water in their homes (S2 Table) and that in fact, households with public network water supply were associated with less probability to belong to burden clusters (S5 Table: PR 0·44; 95%CI 0·21–0·88; p = 0·022). Regarding climate characteristics, these populations share a warm sub-humid climate [18, 19].

In terms of exposure to DENV, most of the participants in each house were seropositive (S2 Table: median 100%; IQR 80–100), and the overall seroprevalence in the population during follow-up increases from 86·9% in 2014 to 90·5% in 2016 [21]. It could be said that these populations are vulnerable from a socio-economic point of view, with a tropical climate and very high exposure to DENV. Although the results presented here represent only these towns, the localities with the highest historical incidence of dengue in Mexico share some characteristics with those studied here. However, some of them are larger and more densely populated [32, 33]. We believe that, regardless of the particularities of each endemic town, the main findings of this work can be applied in dengue epidemiological surveillance systems to rationalize vector control measures.

Even though DENV transmission in the town is heterogeneous, we identified burden clusters that repeat over time or are associated with adjacent clusters, which would allow the

implementation of prevention measures in specific areas of both towns at specific times of the year. Knowledge of the climatic factors associated with recent DENV infection clusters is a necessity both for the environment and for health policy planning, particularly in urban areas. This case aggregation could be due to micro-climatic factors favoring transmission [9] and the accumulation of individuals with a neutralizing immune response against DENV [34]. For example, in previous research conducted by our group in this study area, we determined that the neutralizing multitypic response is predominant from 10 years of age onwards [22]. Furthermore, we determined the seroprevalence against DENV (indirect IgG, Panbio, that assesses having had at least one DENV infection at any time in life) was 90·5% in 2016 [21]. Together, these two findings explain why no association was found between seroprevalence and burden clusters of recent infection DENV.

Dengue transmission is determined by human factors, vector biology and climate [35]; these interact in a complex manner resulting in heterogeneous patterns of transmission [36]. An additional element that introduces heterogeneity is the scale of observation when analyzing the interactions between the aforementioned factors. Steven Stoddard and collaborators emphasize the fact that chains of transmission are established at a minimum neighborhood level, while Henrik Salje and collaborators have determined for dengue transmission in Thailand that 60% of dengue cases living within 200 m distance come from the same chain of transmission, compared to 3% of cases between 1 and 5 km; using a different approach we determined that the risk of infection frequency decreases about by half between the house of an index case and neighboring houses within 50-m radius [5, 8, 33].

In general, in dengue endemic communities, climatic conditions can vary appreciably depending on the constructed area, vegetation and the way water is managed in the area, for example, in this study we found differences of up to 10°C in polygons (S14 and S16 Figs). According to Winberly and collaborators, the areas of highest risk for dengue transmission are those surrounded by impervious areas, adjacent to densely treed areas with high population densities [37]. In this study we demonstrate that areas of higher humidity and the diminished LST are associated with the infection incidence and recurrence over time of dengue cases (Table 2). Considering that chains of transmission are established in discrete areas of the town and that climatic variations are associated with incidence, it can be hypothesized that the heterogeneity of dengue transmission is determined by local changes in population density, urbanization, and the impact that these two factors have on climate at the sub-neighborhood level.

The towns studied have interface characteristics between the drier north of the country with a predominantly dry grassland to desert vegetation and the humid south of Mexico with a very diverse vegetation that includes jungle and extensive wetlands. Nevertheless, the seroprevalence against DENV in the study towns, which are located at the center of Mexico, is approximately 40% in the population aged 6 to 17 years, which is similar to the northern region of the country in this age group; while in the south-southeast of the country the seroprevalence against DENV in the same age group is just over 70% [5]. These differences underline the role of climate in dengue transmission, but also explain why the north of the country has not developed a hyperendemic state as the south of the country clearly has.

As Lessler and collaborators point out, the concept of a transmission cluster or hotspot is used at various spatial and temporal scales and the differences in the data from which clusters are constructed have consequences for the mechanics of transmission and hence the information that is generated for decision-makers [38]. Clustering of recent DENV infections has been reported in the literature. It supports the hypothesis that the incidence of mosquito-borne diseases is highly focal [39], which is possibly influenced by different local densities of *Aedes aegypti* and the short-distance flight of the vector. The latter is consistent with the results of

Acharya and collaborators, who mapped the district-level spatiotemporal distribution of dengue incidence and excess risk in Nepal between 2010 and 2014, finding that the distribution of dengue in that population has significant spatiotemporal clustering [40]. However, similar studies have found a heterogeneous pattern of disease transmission; for example, Sharif and collaborators found no significant associations between dengue case densities and densities of immature forms of the vector in Dhaka city [41].

In addition, it has been observed that at the local spatial level, associations between transmission and the different factors vary from place to place [34, 39, 42–46]. This study identified associations of two microclimatic variables with clusters of recent DENV infection in two Mexican endemic towns. In this regard, the influence of meteorological factors such as temperature, precipitation, and humidity on vector development and survival, vector density, and mosquito oviposition rate has been observed. These factors have also been associated with DENV infection [47]. However, a limitation observed in previous studies is that the resolution of the climate or humidity image is meager [48].

Although DENV circulates throughout the year in tropical and subtropical areas, dengue transmission is highly correlated with temperature, rainy seasons, and seasonal fluctuation of the vector. This relationship between dengue transmission and climatic factors has been previously evaluated in Mexico. For example, Hurtado and collaborators demonstrated that increases in sea surface temperature, minimum temperature, and precipitation were associated with increased dengue transmission cycles in coastal municipalities in the Gulf of Mexico [49]. Similarly, Dzul-Mancilla and collaborators have recently studied transmission cluster distribution in Mexico using aggregated epidemiological surveillance data [50]. The results are robust to the country scale and 8-year time series. However, a significant limitation is that they do not include climate information that could be used to stratify transmission areas using a GIS interface such as the one used for vector surveillance in Mexico [14]. In this study, we applied a method that allowed us to increase the resolution of the LST and soil humidity analysis to 30 m, which is considered a very high resolution. Therefore, it is relatively simple to use this methodology to stratify risk regions, which is one of the strengths of this study.

This study identified recent DENV infections because subjects were followed up and serologically tested every six months. It is not common to have this type of information that allows evidence of the actual burden of the virus and the magnitude of endemic DENV transmission through the detection of both symptomatic and asymptomatic infections. This is the first study in Mexico to use follow-up data from a population cohort to assess DENV transmission and calculate climatic variables with satellite imagery with a high-resolution scale. There is a high frequency of asymptomatic infections in these towns, representing about 60% of the cases [5, 51] and a significant under-reporting of symptomatic infections [52, 53]. This is another strength, as the analysis of only cases reported to the surveillance system underestimates the incidence of infection, making it challenging to identify areas at high risk of transmission within cities [40]. Despite the above, unfortunately, as Brady and collaborators pointed out, the ability to predict dengue outbreaks is inherently unlikely [54]. However, platforms established by the Mexican health system, together with open collaborative information systems and sub-local meteorological information, could streamline vector control activities in a preventive manner, focusing on areas with microenvironmental determinants associated with burden clusters of dengue.

WHO has emphasized the importance of vector surveillance in a comprehensive framework that includes climatic, socioeconomic/cultural and, of course, epidemiological factors where community involvement is important [7, 55]. Efforts have been made to include data science in the analysis of dengue behavior by Google Dengue Trends [56]. However, those who have the operational capacity to interrupt transmission in a preventive manner at the

community scale are ministries of health that do not necessarily possess integrated real-time information platforms with sufficient elements to assess the specific risks of an endemic population. This study allows for a better understanding of the spatiotemporal patterns of DENV burden in endemic Mexican towns for dengue and contributes to the identification of areas and periods of time where the disease occurs to intensify surveillance and event control measures. Another strength of the spatial analysis is the regular and similar shape of both towns, which allowed the construction of a symmetrical polygon grid, respecting geographical boundaries. In addition, using novel sources of information to estimate microclimates within towns allowed us to evaluate these factors for the first time in Mexico.

A limitation of the study is the lack of processing of slightly less than 50% of the cohort samples, since the processing was defined by the objective for the second phase of the study primary in which the present research was nested, the young people were higher represented in the cohort. Furthermore, the study design was based on a population-cohort defined by the dengue case reported to the surveillance system in 2011 rather than a spatial cohort with random sampling. Another possible limitation is that we could detect 8 infections on survey 4; however, there is a small number of infections; it was enough to determine one cluster with neighbor polygons in Tepalcingo. In conclusion, the spread of dengue over time and space is complex and justifies investigations on different spatial scales over extended periods and includes not only cases detected by passive surveillance [57]. Therefore, we consider that this analysis provides insights into burden clusters of recent DENV infection associated with microclimatic factors such as soil humidity and LST, which directly affect the burden of infection. This highlights the importance of using information from alternative sources to determine microclimatic variations in towns to target dengue control measures.

## Supporting information

**S1 Fig. Geographical location of the localities of Axochiapan and Tepalcingo—state of Morelos, Mexico.**
(PDF)

**S2 Fig. Description of follow-up and recent DENV infection.**
(PDF)

**S3 Fig. Spatial distribution and clusters of recent DENV infections in Axochiapan, second survey.** A. Spatial distribution of the participants (Red dots: Positive for recent DENV infection; Green dots: Negative for recent DENV infection). B. Percentage of recent DENV infection. C. Hot Spots of recent DENV infection. D. Clusters (Autocorrelation, Anselin local Moran's I). Sources: Esri module of ArcGIS, DigitalGlobe, GeoEye, Earthstar Geographics, CNES/Airbus DS, USDA, USGS, AeroGRID, IGN, and the GIS User Community.
(PDF)

**S4 Fig. Spatial distribution and clusters of recent DENV infections in Axochiapan, fourth survey.** A. Spatial distribution of the participants (Red dots: Positive for recent DENV infection; Green dots: Negative for recent DENV infection). B. Percentage of recent DENV infection. C. Hot Spots of recent DENV infection. D. Clusters (Autocorrelation, Anselin local Moran's I). Sources: Esri module of ArcGIS, DigitalGlobe, GeoEye, Earthstar Geographics, CNES/Airbus DS, USDA, USGS, AeroGRID, IGN, and the GIS User Community.
(PDF)

**S5 Fig. Spatial distribution and clusters of recent DENV infections in Axochiapan, fifth survey.** A. Spatial distribution of the participants (Red dots: Positive for recent DENV

infection; Green dots: Negative for recent DENV infection). B. Percentage of recent DENV infection. C. Hot Spots of recent DENV infection. D. Clusters (Autocorrelation, Anselin local Moran's I). Sources: Esri module of ArcGIS, DigitalGlobe, GeoEye, Earthstar Geographics, CNES/Airbus DS, USDA, USGS, AeroGRID, IGN, and the GIS User Community.
(PDF)

**S6 Fig. Spatial distribution and clusters of recent DENV infections in Tepalcingo, first survey.** A. Spatial distribution of the participants (Red dots: Positive for recent DENV infection; Green dots: Negative for recent DENV infection). B. Percentage of recent DENV infection. C. Hot Spots of recent DENV infection. D. Clusters (Autocorrelation, Anselin local Moran's I). Sources: Esri module of ArcGIS, DigitalGlobe, GeoEye, Earthstar Geographics, CNES/Airbus DS, USDA, USGS, AeroGRID, IGN, and the GIS User Community.
(PDF)

**S7 Fig. Spatial distribution and clusters of recent DENV infections in Tepalcingo, second survey.** A. Spatial distribution of the participants (Red dots: Positive for recent DENV infection; Green dots: Negative for recent DENV infection). B. Percentage of recent DENV infection. C. Hot Spots of recent DENV infection. D. Clusters (Autocorrelation, Anselin local Moran's I).Sources: Esri module of ArcGIS, DigitalGlobe, GeoEye, Earthstar Geographics, CNES/Airbus DS, USDA, USGS, AeroGRID, IGN, and the GIS User Community.
(PDF)

**S8 Fig. Spatial distribution and clusters of recent DENV infections in Tepalcingo, third survey.** A. Spatial distribution of the participants (Red dots: Positive for recent DENV infection; Green dots: Negative for recent DENV infection). B. Percentage of recent DENV infection. C. Hot Spots of recent DENV infection. D. Clusters (Autocorrelation, Anselin local Moran's I). Sources: Esri module of ArcGIS, DigitalGlobe, GeoEye, Earthstar Geographics, CNES/Airbus DS, USDA, USGS, AeroGRID, IGN, and the GIS User Community.
(PDF)

**S9 Fig. Spatial distribution and clusters of recent DENV infections in Tepalcingo, fourth survey.** A. Spatial distribution of the participants (Red dots: Positive for recent DENV infection; Green dots: Negative for recent DENV infection). B. Percentage of recent DENV infection. C. Hot Spots of recent DENV infection. D. Clusters (Autocorrelation, Anselin local Moran's I). Sources: Esri module of ArcGIS, DigitalGlobe, GeoEye, Earthstar Geographics, CNES/Airbus DS, USDA, USGS, AeroGRID, IGN, and the GIS User Community.
(PDF)

**S10 Fig. Spatial distribution and clusters of recent DENV infections in Tepalcingo, fifth survey.** A. Spatial distribution of the participants (Red dots: Positive for recent DENV infection; Green dots: Negative for recent DENV infection). B. Percentage of recent DENV infection. C. Hot Spots of recent DENV infection. D. Clusters (Autocorrelation, Anselin local Moran's I). Sources: Esri module of ArcGIS, DigitalGlobe, GeoEye, Earthstar Geographics, CNES/Airbus DS, USDA, USGS, AeroGRID, IGN, and the GIS User Community.
(PDF)

**S11 Fig. Location of the repetitive and significant clusters in Axochiapan.**
(PDF)

**S12 Fig. Location of the repetitive and significant clusters in Tepalcingo.**
(PDF)

**S13 Fig. Spatial distribution of soil humidity in Axochiapan by survey.** Red dots: Positive for recent DENV infection; Green dots: Negative for recent DENV infection. A. Survey 1. B. Survey 2. C. Survey 3. D. Survey 4. E. Survey 5. Sources: Esri module of ArcGIS, DigitalGlobe, GeoEye, Earthstar Geographics, CNES/Airbus DS, USDA, USGS, AeroGRID, IGN, and the GIS User Community.
(PDF)

**S14 Fig. Spatial distribution of land surface temperature in Axochiapan by survey.** Blue dots: Positive for recent DENV infection; Green dots: Negative for recent DENV infection. A. Survey 1. B. Survey 2. C. Survey 3. D. Survey 4. E. Survey 5. Sources: Esri module of ArcGIS, DigitalGlobe, GeoEye, Earthstar Geographics, CNES/Airbus DS, USDA, USGS, AeroGRID, IGN, and the GIS User Community.
(PDF)

**S15 Fig. Spatial distribution of soil humidity in Tepalcingo by survey.** Red dots: Positive for recent DENV infection; Green dots: Negative for recent DENV infection. Black dots: Non-included.A. Survey 1. B. Survey 2. C. Survey 3. D. Survey 4. E. Survey 5. Sources: Esri module of ArcGIS, DigitalGlobe, GeoEye, Earthstar Geographics, CNES/Airbus DS, USDA, USGS, AeroGRID, IGN, and the GIS User Community.
(PDF)

**S16 Fig. Spatial distribution of land surface temperature in Tepalcingo by survey.** Red dots: Positive for recent DENV infection; Green dots: Negative for recent DENV infection. Black dots: Non-included. A. Survey 1. B. Survey 2. C. Survey 3. D. Survey 4. E. Survey 5. Sources: Esri module of ArcGIS, DigitalGlobe, GeoEye, Earthstar Geographics, CNES/Airbus DS, USDA, USGS, AeroGRID, IGN, and the GIS User Community.
(PDF)

**S1 Table. Sociodemographic characteristics of the study population by survey.**
(PDF)

**S2 Table. Characteristics of the houses by survey.**
(PDF)

**S3 Table. Significant clusters of recent DENV infection in Axochiapan.**
(PDF)

**S4 Table. Significant clusters of recent DENV infection in Tepalcingo.**
(PDF)

**S5 Table. Housing characteristics related to DENV burden clusters, univariable analysis.**
*Poor housing maintenance: houses were those having a poor structure, poor organization, dirty, with unpainted or cracked walls, improvised sections, or broken windows/doors
**House with larvae or pupae of *Aedes*.
(PDF)

## Acknowledgments

The authors of this paper gratefully acknowledge the comments of colleagues from the Peer Editing Program of the International Society for Environmental Epidemiology (https://iseepi. org/) on this manuscript.

## Author Contributions

**Conceptualization:** Ruth Aralí Martínez-Vega, José Ramos-Castañeda.

**Data curation:** Susana Román-Pérez, Irma Yvonne Amaya-Larios.

**Formal analysis:** Johanna Tapias-Rivera, Ruth Aralí Martínez-Vega, Susana Román-Pérez.

**Funding acquisition:** José Ramos-Castañeda.

**Methodology:** Ruth Aralí Martínez-Vega, Rene Santos-Luna, Fredi Alexander Diaz-Quijano, José Ramos-Castañeda.

**Project administration:** Irma Yvonne Amaya-Larios.

**Writing – original draft:** Johanna Tapias-Rivera, Ruth Aralí Martínez-Vega, José Ramos-Castañeda.

**Writing – review & editing:** Johanna Tapias-Rivera, Ruth Aralí Martínez-Vega, Susana Román-Pérez, Rene Santos-Luna, Irma Yvonne Amaya-Larios, Fredi Alexander Diaz-Quijano, José Ramos-Castañeda.

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
