## [Decision Letter · Decision Letter 0]

3 Dec 2023

PONE-D-23-34212Microclimate factors related to dengue virus burden clusters in two endemic towns of MexicoPLOS ONE

Dear Dr. Ramos-Castañeda,

Thank you for submitting your manuscript to PLOS ONE. After careful consideration, we feel that it has merit but does not fully meet PLOS ONE’s publication criteria as it currently stands. Therefore, we invite you to submit a revised version of the manuscript that addresses the points raised during the review process.

We look forward to receiving your revised manuscript.

Kind regards,

Kovy Arteaga-Livias

Academic Editor

PLOS ONE

“FAD-Q declares grants from the Brazilian National Council for Scientific and Technological Development – CNPq and Sanofi Pasteur not related to this work. JR-C declares grant support to develop the cohort from which data are taken and funded by Sanofi Pasteur DNG 22 (Dengue seroprevalence, neutralizing titers and incidence in an endemic population of Morelos State, Mexico); payment for expert testimony from Takeda and support for attending meetings and/or travel by Sanofi Pasteur not related to this work. RAM-V reports honoraria and travel expenses as a consultant for the project from which the primary data were obtained by the National Institute of Public Health. Mexico and funded by Sanofi Pasteur DGN 22. IYA-L reports honoraria as coordinator of the project from which the primary data were obtained by the National Institute of Public Health. Mexico and funded by Sanofi Pasteur DNG 22. JT-P, SR-P and RS-L have no conflict of interest to declare.”

4. We note that Figures 1,2,3,S1,S3,S4,S5,S6,S7,S8,S9,S10,S11 and S12 in your submission contain [map/satellite] images which may be copyrighted. All PLOS content is published under the Creative Commons Attribution License (CC BY 4.0), which means that the manuscript, images, and Supporting Information files will be freely available online, and any third party is permitted to access, download, copy, distribute, and use these materials in any way, even commercially, with proper attribution. For these reasons, we cannot publish previously copyrighted maps or satellite images created using proprietary data, such as Google software (Google Maps, Street View, and Earth). For more information, see our copyright guidelines: http://journals.plos.org/plosone/s/licenses-and-copyright.

a. You may seek permission from the original copyright holder of Figures 1,2,3,S1,S3,S4,S5,S6,S7,S8,S9,S10,S11 and S12 to publish the content specifically under the CC BY 4.0 license. 

Reviewers' comments:

Reviewer's Responses to Questions

**Comments to the Author**

1. Is the manuscript technically sound, and do the data support the conclusions?

Reviewer #1: Partly

Reviewer #2: Partly

Reviewer #3: Yes

2. Has the statistical analysis been performed appropriately and rigorously? 

Reviewer #1: N/A

Reviewer #2: I Don't Know

Reviewer #3: Yes

3. Have the authors made all data underlying the findings in their manuscript fully available?

Reviewer #1: Yes

Reviewer #2: Yes

Reviewer #3: Yes

4. Is the manuscript presented in an intelligible fashion and written in standard English?

Reviewer #1: Yes

Reviewer #2: Yes

Reviewer #3: Yes

5. Review Comments to the Author

Reviewer #1: The study is really interesting but some limitations are present.

1. How did the authors select two cities ? Please describe the criteria for choosing those selected two study areas in this study at the revised manuscript.

2. Panbio ELISA is the gold standard test but PBU 9-11 was equivocal result. How did the authors categorize for the equivocal results samples? Please describe clearly in the methodology section.

3.Anti-DENV IgM Ab can use for determining recent DENV infection. Please add the operational definitions for defining recent DENV infection in this study at the revised manuscript. What is the purpose for anti-DENV IgG ELISA? Please discuss and describe the justification for doing this one.

4.DENV is one of the members of flaviviruses and how about the cross reactivity with other flaviviruses if the authors use only ELISA test for determining DENV infection. there is cross reactivity with other flaviviruses. We did not see any information about the cross reactivity of the tests for determining the recent DENV infection. Please briefly discuss about the limitation of the test used in this study.

5.Why did not the authors choose NS-1 Ag tests which is available in RDT kit? If NS-1 Ag is positive, we can say very strongly that DENV infection is active and transmitted. Please discuss briefly on this issue.

6. The authors described that many hotspots were identified in the study. But the authors choose IgM Ab for determing recent DENV infection. The IgM Ab can persists for three months. How can the authors say that the infected persons within hotspot got infection at the same period? Eg, In a hotspot one case got infection last three months ago and one case got infection at the time of travelling last one month. Although those houses of two cases were very close, we could not say this is active transmission of DENV at that hotspot. Only PCR can say and we can say same serotype or not/ Please also discuss on this issue.This is the disadvantages of using Ab only in this study. Please add as limitations of the study at the revised manuscript.

Reviewer #2: The work by Tapias-Rivera and colleagues describes the analysis de microclimate factors related to dengue virus burden clusters in two endemic towns of Mexico. For the analysis they used collected data from previous studies performed during 2014 to 2016. The authors aimed to identify if dengue burden clusters are associated or related with two climate variables, LST and soil humidity.

The work is interesting and brings to discussion the use of clusters and hotspots as a methodological strategy to inform health authorities on high transmission zones.

However, there´s some concerns about the data presented by the authors.

1. There´s not clarity of the period of the study until the methods section. It should be clarified that data analysis is retrospective and not a “prospective population-based cohort study” as they mention on page 6 line 102. Thus, when authors mention “recent DENV infections” (although they define it) those are not related to current or new data. The word recent should be removed from all sections or tables or figures, it causes confusion. The study uses data collected from a decade ago.

2. There´s not clarity on the data use and analysis when refer to clusters and hotspots. According to the source of information section, data are divided in those from people >5 years old collected during dengue transmission and those from the seroconversion dynamics study. But the analysis was performed with a section of the data, from participants who were assessed in five occasions. Then, its not clear why the analysis for each town is different. In the hotspots they described for participants from Axiochapan, results for surveys 1 and 3 but not mention what happen in the remaining surveys. However, for Tepalcingo, they described results for all five surveys. A similar description is found on the Microclimate analysis. If the data to perform the analysis its not significant for all five surveys for each town, it should be mentioned.

3. The numbers presented in S2 Figure are the totals for both towns. But the authors did not mention the numbers in each town. How significant is to perform the hotspots and microclimate analysis with a small number of cases, with 88 the highest and 8 the lowest in each survey? This should be mentioned in the results section and discussed.

4. Data discussed in page 16, lines 314 to 318 are overinterpreted. The author’s comparison is with data reported in 2003. How data that were collected a decade ago by the authors can be applied during the actual dengue epidemiological surveillance system. How factors such as population mobility, dengue seroprevalence, dengue outbreaks since 2014-2016 could affect what happened ten years ago? Although the burden clusters repeated for two years (2014-2016), before making any assumption, they should be compared with the actual situation.

Reviewer #3: 1.Definition of Recent DENV infection is not clear. The author want to say DENV IgM positive or DENV IgM and IgG positive?Please calrify it.

2. The author did 5 surveys. Survey 1 showed the highest recent DENV infection. Why? Survey 1 period (Aug-Nov 2014) is big dengue outbreak in that area or other reasons?

3. Based on Survey 1-5, 2014 the highest % recent infection, 2015 lowest % recent infection and 2016 recent infection higher than 2015, but lower than 2014. It is better to discuss recent infection trend condition in the discussion.

4. land surface temperature, soil humidity were not independently associated with DENV burden clusters. So, which possible factors are associated with DENV burden? please discuss it.

5.Significant clusters of recent DENV infection described in Axochiapan (S4) and Tepalcingo (S5). Recent infection % are different on Survey period. But it is clearly to explain in result part.

6. Supplementary data are too much. It is better to omit some figures and tables.

6. PLOS authors have the option to publish the peer review history of their article (what does this mean?). If published, this will include your full peer review and any attached files.

Reviewer #1: No

Reviewer #2: No

Reviewer #3: No

---

## [Author Response · Author response to Decision Letter 0]

15 Jan 2024

Reviewers' comments:

Comments to the Author

Reviewer #1

The study is really interesting but some limitations are present.

1. How did the authors select two cities ? Please describe the criteria for choosing those selected two study areas in this study at the revised manuscript.

RESPONSE: We thank the reviewer for the comment. We accepted the suggestions and modify the paragraph in the section materials and methods. The new paragraph says: “Two dengue endemic towns in the state of Morelos, Mexico was selected taking into account the epidemiological information available up to the time of the study, which was provided by the Ministry of Health (SSA) of the federal government, where it was established that Morelos was the state with the most confirmed cases of dengue in Mexico. The first of them, Axochiapan (the urban area of the municipality of the same name) has approximately 18,659 inhabitants according to the 2015 INEGI estimated population; the temperature ranges between 13 °C and 35 °C. The second one, Tepalcingo (the urban area of the municipality of the same name) has approximately 12,895 inhabitants; the temperature varies between 19 °C and 34 °C (S1 Fig) [17-19].

Sources of information: A secondary analysis was conducted on data collected in a prospective population-based cohort study of people aged five years and older living in Tepalcingo and Axochiapan between 2014 and 2016. This cohort was assembled in 2011 and the objective of the primary study was to determine that dengue transmission occurs primarily in the peridomestic area of Index Cases (IC) in two Mexican endemic towns [5], which were selected because they were the localities in Morelos with rates of particularly high incidence of dengue, with similar population density and where at that time the vector surveillance program was being carried out by the Ministry of Health since 2008.”

2. Panbio ELISA is the gold standard test but PBU 9-11 was equivocal result. How did the authors categorize for the equivocal results samples? Please describe clearly in the methodology section.

RESPONSE: We included Table 1 in the main text to explain the definitions of infection according to the stages of the study. In general, indeterminates were considered negative using all three techniques. In addition, the proportion of indeterminate results for each test was calculated to estimate their impact on the number of infections.

3.Anti-DENV IgM Ab can use for determining recent DENV infection. Please add the operational definitions for defining recent DENV infection in this study at the revised manuscript. What is the purpose for anti-DENV IgG ELISA? Please discuss and describe the justification for doing this one.

RESPONSE: In this sense, as mentioned in the previous point, Table 1 was added (please see the previous answer). On the other hand, for the capture of IgG, the Mexican algorithm for diagnosing Zika virus transmission considers the capture of IgG as confirmation of dengue infection since, in Mexico, there was no other flavivirus circulating until November 2015 when Zika virus started to circulate.

4.DENV is one of the members of flaviviruses and how about the cross reactivity with other flaviviruses if the authors use only ELISA test for determining DENV infection. there is cross reactivity with other flaviviruses. We did not see any information about the cross reactivity of the tests for determining the recent DENV infection. Please briefly discuss about the limitation of the test used in this study.

RESPONSE: As mentioned in the previous answer, in Mexico and specifically in the study area, only the Zika virus started circulating in November 2015. As can be seen in Table 1, in surveys 3, 4 and 5, the result of IgG capture was excluded from the definition of recent infection, as the result of this test is particularly influenced by the cross-reactivity of flaviviruses; only those positive for IgM were considered DENV infections, as this test has a specificity greater than 90% using any diagnostic kit; the other criterion was seroconversion as measured by the previous paired sample in an indirect IgG assay that measures DENV exposure in historical samples; in this case, a negative result in the previous sample together with a positive result, even in the absence of positive IgM, is evidence of recent infection.

This is explained in the text as with Table 1 of the manuscript.

5.Why did not the authors choose NS-1 Ag tests which is available in RDT kit? If NS-1 Ag is positive, we can say very strongly that DENV infection is active and transmitted. Please discuss briefly on this issue.

RESPONSE: We disagree with the reviewer as this study aimed to detect total infections, not only symptomatic infections, which is the indication for the use of NS1 determination.

6. The authors described that many hotspots were identified in the study. But the authors choose IgM Ab for determing recent DENV infection. The IgM Ab can persists for three months. How can the authors say that the infected persons within hotspot got infection at the same period? Eg, In a hotspot one case got infection last three months ago and one case got infection at the time of travelling last one month. Although those houses of two cases were very close, we could not say this is active transmission of DENV at that hotspot. Only PCR can say and we can say same serotype or not/ Please also discuss on this issue.This is the disadvantages of using Ab only in this study. Please add as limitations of the study at the revised manuscript.

RESPONSE: We respectfully disagree with the reviewer. As explained in the previous answer, the project aimed to determine total infections, so we clearly state that we are studying recent infections, not active infections, as the reviewer suggests. This nuance is relevant because we intend to study transmission dynamics in the context of local climatic conditions, not disease incidence. Furthermore, we have evidence in these localities the dengue transmission mainly is peri domiciliary (Martínez-Vega RA, et al. PLoS Negl Trop Dis. 2015 Dec 15;9(12):e0004296.), considering that mobility outside the community is not significant (Falcón-Lezama JA, et al. PLoS One. 2017 Feb 22;12(2):e0172313.)

Reviewer #2

The work by Tapias-Rivera and colleagues describes the analysis de microclimate factors related to dengue virus burden clusters in two endemic towns of Mexico. For the analysis they used collected data from previous studies performed during 2014 to 2016. The authors aimed to identify if dengue burden clusters are associated or related with two climate variables, LST and soil humidity.

The work is interesting and brings to discussion the use of clusters and hotspots as a methodological strategy to inform health authorities on high transmission zones.

However, there´s some concerns about the data presented by the authors.

1. There´s not clarity of the period of the study until the methods section.

RESPONSE: We thank the reviewer for the comment. We clarified the paragraph in the Introduction section to include the study period where the data were collected. The new phrase says: “Therefore, the study aimed to assess the association between some microenvironmental determinants and the occurrence of burden clusters of DENV infection in a population-based cohort in two endemic towns in southern Mexico from 2014 to 2016”.

 It should be clarified that data analysis is retrospective and not a “prospective population-based cohort study” as they mention on page 6 line 102. 

RESPONSE: We are sorry, but we do not agree with the reviewer. The study group considers that the work corresponds to a prospective cohort study because the information collection process was prospective, including the consent process and the blood draw. However, the present manuscript reports the results of a secondary analysis, as this was not a specific objective of the primary project.

Thus, when authors mention “recent DENV infections” (although they define it) those are not related to current or new data. The word recent should be removed from all sections or tables or figures, it causes confusion. The study uses data collected from a decade ago.

RESPONSE: We understand the reviewer's comment, however, when the recent infection is mentioned, it refers to the fact that those were recent infections when the samples were processed for the study period analyzed. We don´t want confusion with a seroprevalence study therefore we use “recent infection” for clarity.

2. There´s not clarity on the data use and analysis when refer to clusters and hotspots. According to the source of information section, data are divided in those from people >5 years old collected during dengue transmission and those from the seroconversion dynamics study. But the analysis was performed with a section of the data, from participants who were assessed in five occasions. Then, its not clear why the analysis for each town is different. In the hotspots they described for participants from Axiochapan, results for surveys 1 and 3 but not mention what happen in the remaining surveys. However, for Tepalcingo, they described results for all five surveys. A similar description is found on the Microclimate analysis. If the data to perform the analysis its not significant for all five surveys for each town, it should be mentioned.

RESPONSE: We thank the reviewer for the comment, and we modified what was suggested in order to clarify the evaluation process during the 2.5 years of survey was the same for both localities; the new paragraph says:

“In Axochiapan, in surveys 1, 3, and 5 (August-November 2014, 2015, and 2016), 32 hotspots of recent DENV infection were observed predominantly in the southeastern area of the town. In Axochiapan surveys 2 and 4 (February-May 2015 and 2016), 14 hotspots were observed. However, autocorrelation analysis only identified significant clusters of recent DENV infection in six polygons in three neighborhoods in survey 1 (Vista Hermosa to the northwest and El Carmen and El Progreso to the southeast, with infection frequencies between 10% and 100%) and four polygons of two neighborhoods in survey 3 (El Carmen and El Progreso with infection frequencies between 5·6% and 33·3%) (Figs 1 and 2, S4 Table). The hotspots identified in surveys 2, 4, and 5 in this town did not consolidate as clusters of the burden of DENV infection.”

Also, we add S3-S5 representing Axochiapan survey 2, 4 and 5 on the supplementary information file.

3. The numbers presented in S2 Figure are the totals for both towns. But the authors did not mention the numbers in each town. How significant is to perform the hotspots and microclimate analysis with a small number of cases, with 88 the highest and 8 the lowest in each survey? This should be mentioned in the results section and discussed.

RESPONSE: Figure S2 was modified to show the number of infections in each location. Furthermore, as the reviewer mentions, the number of infections is small, but as we find statistically significant results with this number of infections, the effect of climatic variables on dengue transmission should be even more significant than this study suggests. Therefore, we modified the Discussion section and added the following text to the limitations paragraph:

"Another possible limitation is that we were able to detect 8 infections in survey 4; However, there is a small number of infections; It was enough to determine a cluster with neighboring polygons in Tepalcingo"

4. Data discussed in page 16, lines 314 to 318 are overinterpreted. The author’s comparison is with data reported in 2003. How data that were collected a decade ago by the authors can be applied during the actual dengue epidemiological surveillance system. How factors such as population mobility, dengue seroprevalence, dengue outbreaks since 2014-2016 could affect what happened ten years ago? Although the burden clusters repeated for two years (2014-2016), before making any assumption, they should be compared with the actual situation.

RESPONSE: First, we would like to point out that the paragraph mentioned by the reviewer does not refer to any specific moment in time, so we do not know what the reviewer means by compare data from 2003. The reviewer mentions that the results are overinterpreted as the retrospective information does not allow comparison with the current situation, however, this report intends to describe how environmental variables influence the peridomiciliary transmission of DENV and thus eventually improve surveillance systems that currently, at best, only consider incidence and entomological data. In other words, this study does not intend to compare the dengue situation in these localities in 2003 with the current situation, so we disagree with the reviewer's comment.

Reviewer #3

 1.Definition of Recent DENV infection is not clear. The author want to say DENV IgM positive or DENV IgM and IgG positive?Please calrify it.

RESPONSE: We appreciate the suggestion and revised and included Table 1 in the main text to explain the definitions of infection according to the stages of the study. 

2. The author did 5 surveys. Survey 1 showed the highest recent DENV infection. Why? Survey 1 period (Aug-Nov 2014) is big dengue outbreak in that area or other reasons?

RESPONSE: The study group considers this situation because, according to the Epidemiological Panorama of dengue, in 2014, there were more cases in Morelos than in the following two years. In particular, a locality (Tepalcingo) was reported in the state's second-highest number of cases. (https://www.gob.mx/salud/documentos/direction-general-de-epidemiologia-panorama-epidemiologico-de-dengue -2014-semana-epidemiologica-53; https://www.gob.mx/salud/documentos/direction-general-de-epidemiologia-panorama-epidemiologico-de-dengue-2016-semana-epidemiologica-52)

Accordingly, we add the following text: “Moreover, in 2014, more dengue cases were reported by the surveillance system in Morelos than in the following two years. In particular, Tepalcingo had the state's second-highest number of cases.”

3. Based on Survey 1-5, 2014 the highest % recent infection, 2015 lowest % recent infection and 2016 recent infection higher than 2015, but lower than 2014. It is better to discuss recent infection trend condition in the discussion.

We thank you and accept the recommendation. In the discussion section, we include a text about this. The new sentence says: "with the known cyclical incidence rate described in many studies in endemic cities" ( Wearing HJ, Rohani P. Ecological and immunological determinants of dengue epidemics Proc Natl Acad Sci U S A. 2006 ;103(31):11802-7 and Adams B, Holmes EC, Zhang C, Mammen MP, Jr., Nimmannitya S, Kalayanarooj S, et al. alternating epidemic pattern of dengue virus serotypes circulating in Bangkok, Proc Natl Acad Sci U S A. 2006;103(38):14234-9).

4. land surface temperature, soil humidity were not independently associated with DENV burden clusters. So, which possible factors are associated with DENV burden? please discuss it.

RESPONSE: We are confused. As mentioned in the results section, “Hexagonal LST was negatively associated with households belonging to a DENV loading group, while high soil moisture had a positive association with this outcome (Table 2). Therefore, that is exactly the observation, that climatic factors determine the burden clusters.

5.Significant clusters of recent DENV infection described in Axochiapan (S4) and Tepalcingo (S5). Recent infection % are different on Survey period. But it is clearly to explain in result part.

RESPONSE: We regret that we do not understand what the reviewer is commenting on and therefore cannot give a detailed answer. However, the response to reviewer 2's comment 2 may be associated with this observation.

6. Supplementary data are too much. It is better to omit some figures and tables.

RESPONSE: We appreciate the reviewer's recommendation, however, in the opinion of all the authors, the supplementary information is the minimum sufficient to understand the entire manuscript. Therefore, no further editing was done in this regard.

---

## [Decision Letter · Decision Letter 1]

9 Feb 2024

PONE-D-23-34212R1Microclimate factors related to dengue virus burden clusters in two endemic towns of MexicoPLOS ONE

Dear Dr. Ramos-Castañeda,

Thank you for submitting your manuscript to PLOS ONE. After careful consideration, we feel that it has merit but does not fully meet PLOS ONE’s publication criteria as it currently stands. Therefore, we invite you to submit a revised version of the manuscript that addresses the points raised during the review process.

We look forward to receiving your revised manuscript.

Kind regards,

Kovy Arteaga-Livias

Academic Editor

PLOS ONE

Reviewers' comments:

Reviewer's Responses to Questions

**Comments to the Author**

1. If the authors have adequately addressed your comments raised in a previous round of review and you feel that this manuscript is now acceptable for publication, you may indicate that here to bypass the “Comments to the Author” section, enter your conflict of interest statement in the “Confidential to Editor” section, and submit your "Accept" recommendation.

Reviewer #1: All comments have been addressed

Reviewer #2: (No Response)

Reviewer #3: All comments have been addressed

2. Is the manuscript technically sound, and do the data support the conclusions?

Reviewer #1: Yes

Reviewer #2: Partly

Reviewer #3: Yes

3. Has the statistical analysis been performed appropriately and rigorously? 

Reviewer #1: N/A

Reviewer #2: I Don't Know

Reviewer #3: Yes

4. Have the authors made all data underlying the findings in their manuscript fully available?

Reviewer #1: Yes

Reviewer #2: No

Reviewer #3: Yes

5. Is the manuscript presented in an intelligible fashion and written in standard English?

Reviewer #1: Yes

Reviewer #2: Yes

Reviewer #3: No

6. Review Comments to the Author

Reviewer #1: (No Response)

Reviewer #2: The authors have addressed some of the comments from previous revision. However, there´s still lack of clarity on the source of information section and recent dengue infection section.

It is suggested to the authors to re-write the Source of Information Section. According to what they described, a cohort was first assembled in 2011 (reference 5 from the same team). This cohort included 862 participants. Then, for the secondary study performed during 2014 to 2016, they added 120 participants; although this number doesn't matches the number (982) on the text (page 6 line 116) neither the number on figure S2 (966 participants).

A confusion also comes when they mention in page 6 lines 114-118 the following... This analysis included the subgroup of persons evaluated during the second phase of the cohort study who had participated during the first phase of the cohort and were DENV seronegative or had a DENV infection (n=461 of 862 included between August 2011 and March 2012)[5], or who were recruited in the second phase of the cohort and were DENV seronegative (n=19 of 120 included between August and November 2014) (S2 Fig).

Then, for the five serosurveys, they included different participants from the total of 966? or 982? or 862? Clarify which is the total number of participants in this study.

The section Recent Dengue Infection. The authors explained the reasoning to consider "recent infection" although it was suggested to eliminate this word. They argue the fact that only Dengue was in circulation before Zika introduction. What would be the result after Zika introduction in 2015? they did survey 3 in the second semester of 2015 and surveys 4 and 5 in 2016 when even Chikungunya was introduced in Mexico.

The selection of the two sites for performing the study was according to the authors, because were dengue-endemic localities in Morelos with high incidence rates. However, the authors never mention in the document which was this incidence, nor the number of dengue cases reported by the Health Ministry.

They mention in the Discussion section (page 17, lines 311-312).... Moreover, in 2014, more dengue cases were reported by the surveillance system in Morelos than in the following two years. In particular, Tepalcingo had the state's second-highest number of dengue cases. Include the numbers.

The authors can revise the following article: https://www.medigraphic.com/pdfs/infectologia/lip-2020/lip202d.pdf. In page 80, there´s a summary of confirmed cases of Dengue from 2000 to 2019.

Reviewer #3: The revised manuscript addressed my comments. there has no more comments for me. I accepted revised manuscript.

7. PLOS authors have the option to publish the peer review history of their article (what does this mean?). If published, this will include your full peer review and any attached files.

Reviewer #1: **Yes: **Aung Kyaw Kyaw

Reviewer #2: No

Reviewer #3: No

---

## [Author Response · Author response to Decision Letter 1]

13 Feb 2024

Response to PONE-D-23-34212R1, Microclimate factors related to dengue virus burden clusters in two endemic towns of Mexico. 

On behalf of all the authors I would like to thank the editor and reviewers for their comments. Here we respond to the comments expressed by the reviewers in detail.

Reviewer #1: (No Response)

Reviewer #2: The authors have addressed some of the comments from previous revision. However, there´s still lack of clarity on the source of information section and recent dengue infection section.

It is suggested to the authors to re-write the Source of Information Section. According to what they described, a cohort was first assembled in 2011 (reference 5 from the same team). This cohort included 862 participants. Then, for the secondary study performed during 2014 to 2016, they added 120 participants; although this number doesn't matches the number (982) on the text (page 6 line 116) neither the number on figure S2 (966 participants).

A confusion also comes when they mention in page 6 lines 114-118 the following... This analysis included the subgroup of persons evaluated during the second phase of the cohort study who had participated during the first phase of the cohort and were DENV seronegative or had a DENV infection (n=461 of 862 included between August 2011 and March 2012)[5], or who were recruited in the second phase of the cohort and were DENV seronegative (n=19 of 120 included between August and November 2014) (S2 Fig).

Then, for the five serosurveys, they included different participants from the total of 966? or 982? or 862? Clarify which is the total number of participants in this study.

RESPONSE: We are grateful to the reviewer for bringing this typographical error to our attention and it has been corrected as mentioned below (changes in bold): 

Line 120, page 7: "...DENV seronegative (n=19 of 104 included...).

We also corrected a confusing paragraph as follows:

Line 116-117, page 6: "...had a recent DENV infection, or were randomly selected from seropositive participants without DENV recent infection..."

We hope that with these changes the point made by the reviewer is clarified.

The section Recent Dengue Infection. The authors explained the reasoning to consider "recent infection" although it was suggested to eliminate this word. They argue the fact that only Dengue was in circulation before Zika introduction. What would be the result after Zika introduction in 2015? they did survey 3 in the second semester of 2015 and surveys 4 and 5 in 2016 when even Chikungunya was introduced in Mexico.

RESPONSE: We disagree with the reviewer about removing the definition of recent infection, for two reasons. 

1) The reviewer, without explicitly mentioning it, suggests that due to antigenic cross-reactivity with other flaviviruses a serological definition of recent infection cannot be used. However, if there is no other flavivirus circulating in the area, how could IgM against DENV be positive, or, for that matter, what infection would cause IgG seroconversion in the indirect IgG testing for DENV? The reviewer mentions the case of Chikungunya virus, which has certainly circulated in Mexico since 2014, although transmission in the study state was reported only in 2015 (https://www.gob.mx/salud/acciones-y-programas/historico-boletin-epidemiologico); in any case, Chikunganya virus is not antigenically related to Dengue virus, because it is an alphavirus of the Togaviridae family. Thus, it was only until the report of Zika virus circulation in Mexico in 2015 that it was decided to omit the IgG capture test from the definition of recent infection, because of the possibility of antigenic cross-reaction, but we should mention that only until 2016 was Zika virus circulation reported in Morelos (op cit).

2) In at least three previous publications, Martínez-Vega RA, et al. PLoS Negl Trop Dis. 2015;9(12):e0004296; Falcón-Lezama JA, et al. PLoS One. 2017;12(2):e0172313 and Amaya-Larios IY, et al. Sci Rep. 2020;10(1):19017, we used an identical or very similar definition when describing cases of recent DENV infection in the cohort.

Therefore, we stand by our definition of recent infection as described throughout the manuscript.

The selection of the two sites for performing the study was according to the authors, because were dengue-endemic localities in Morelos with high incidence rates. However, the authors never mention in the document which was this incidence, nor the number of dengue cases reported by the Health Ministry.

They mention in the Discussion section (page 17, lines 311-312).... Moreover, in 2014, more dengue cases were reported by the surveillance system in Morelos than in the following two years. In particular, Tepalcingo had the state's second-highest number of dengue cases. Include the numbers.

The authors can revise the following article: https://www.medigraphic.com/pdfs/infectologia/lip-2020/lip202d.pdf. In page 80, there´s a summary of confirmed cases of Dengue from 2000 to 2019.

RESPONSE: The reference suggested by the reviewer mentions cumulative dengue cases in Mexico, which does not contribute to resolve the point the reviewer mentions; however, to meet the reviewer's observation we add a reference, Martínez-Vega RA, Danis-Lozano R, Velasco-Hernández J, Díaz-Quijano FA, González-Fernández M, Santos R, Román S, Argáez-Sosa J, Nakamura M, Ramos-Castañeda J. A prospective cohort study to evaluate peridomestic infection as a determinant of dengue transmission: protocol. BMC Public Health. 2012 Apr 2;12:262, which details the epidemiological indicators requested by the reviewer, specifically for the two studied localities.

Therefore, the Study area of the Methodology section was modified with the following text: Details of dengue transmission in the selected localities can be found in Martinez-Vega RA, et al BMC Public Health. 2012;12:262.

Reviewer #3: The revised manuscript addressed my comments. there has no more comments for me. I accepted revised manuscript.

José Ramos-Castañeda PhD

---

## [Decision Letter · Decision Letter 2]

27 Mar 2024

Microclimate factors related to dengue virus burden clusters in two endemic towns of Mexico

PONE-D-23-34212R2

Dear Dr. Ramos-Castañeda,

We’re pleased to inform you that your manuscript has been judged scientifically suitable for publication and will be formally accepted for publication once it meets all outstanding technical requirements.

Kind regards,

Kovy Arteaga-Livias

Academic Editor

PLOS ONE

Additional Editor Comments (optional):

Reviewers' comments:

Reviewer's Responses to Questions

**Comments to the Author**

1. If the authors have adequately addressed your comments raised in a previous round of review and you feel that this manuscript is now acceptable for publication, you may indicate that here to bypass the “Comments to the Author” section, enter your conflict of interest statement in the “Confidential to Editor” section, and submit your "Accept" recommendation.

Reviewer #2: All comments have been addressed

2. Is the manuscript technically sound, and do the data support the conclusions?

Reviewer #2: Yes

3. Has the statistical analysis been performed appropriately and rigorously? 

Reviewer #2: Yes

4. Have the authors made all data underlying the findings in their manuscript fully available?

Reviewer #2: Yes

5. Is the manuscript presented in an intelligible fashion and written in standard English?

Reviewer #2: Yes

6. Review Comments to the Author

Reviewer #2: No additional comments. The suggestions have been satisfactory resolved buy the authors. The work is accepted for publication.

7. PLOS authors have the option to publish the peer review history of their article (what does this mean?). If published, this will include your full peer review and any attached files.

Reviewer #2: No

---

## [Editor Report · Acceptance letter]

27 May 2024

PONE-D-23-34212R2 

PLOS ONE

Dear Dr. Ramos-Castañeda, 

I'm pleased to inform you that your manuscript has been deemed suitable for publication in PLOS ONE. Congratulations! Your manuscript is now being handed over to our production team.

Kind regards, 

on behalf of

Dr. Kovy Arteaga-Livias 

Academic Editor

PLOS ONE